# Activation of Nrf2 in Astrocytes Suppressed PD-Like Phenotypes via Antioxidant and Autophagy Pathways in Rat and Drosophila Models

**DOI:** 10.3390/cells10081850

**Published:** 2021-07-21

**Authors:** Qing Guo, Bing Wang, Xiaobo Wang, Wanli W. Smith, Yi Zhu, Zhaohui Liu

**Affiliations:** 1Department of Human Anatomy and Cytoneurobiology, Medical School of Soochow University, Suzhou 215123, China; guoqing@suda.edu.cn (Q.G.); wangbing19881129@126.com (B.W.); wangxiaobo86@139.com (X.W.); 2Department of Psychiatry and Behavioral Sciences, Johns Hopkins University School of Medicine, Baltimore, MD 21287, USA; wsmith60@jhmi.edu; 3Department of Pharmacology, Medical School of Soochow University, Suzhou 215123, China

**Keywords:** Parkinson’s disease, Nrf2, neurodegeneration, oxidative stress, autophagy

## Abstract

The oxidative-stress-induced impairment of autophagy plays a critical role in the pathogenesis of Parkinson’s disease (PD). In this study, we investigated whether the alteration of Nrf2 in astrocytes protected against 6-OHDA (6-hydroxydopamine)- and rotenone-induced PD-like phenotypes, using 6-OHDA-induced rat PD and rotenone-induced Drosophila PD models. In the PD rat model, we found that Nrf2 expression was significantly higher in astrocytes than in neurons. CDDO-Me (CDDO methyl ester, an Nrf2 inducer) administration attenuated PD-like neurodegeneration mainly through Nrf2 activation in astrocytes by activating the antioxidant signaling pathway and enhancing autophagy in the substantia nigra and striatum. In the PD Drosophila model, the overexpression of Nrf2 in glial cells displayed more protective effects than such overexpression in neurons. Increased Nrf2 expression in glial cells significantly reduced oxidative stress and enhanced autophagy in the brain tissue. The administration of the Nrf2 inhibitor ML385 reduced the neuroprotective effect of Nrf2 through the inhibition of the antioxidant signaling pathway and autophagy pathway. The autophagy inhibitor 3-MA partially reduced the neuroprotective effect of Nrf2 through the inhibition of the autophagy pathway, but not the antioxidant signaling pathway. Moreover, Nrf2 knockdown caused neurodegeneration in flies. Treatment with CDDO-Me attenuated the Nrf2-knockdown-induced degeneration in the flies through the activation of the antioxidant signaling pathway and increased autophagy. An autophagy inducer, rapamycin, partially rescued the neurodegeneration in Nrf2-knockdown Drosophila by enhancing autophagy. Our results indicate that the activation of the Nrf2-linked signaling pathways in glial cells plays an important neuroprotective role in PD models. Our findings not only provide a novel insight into the mechanisms of Nrf2–antioxidant–autophagy signaling, but also provide potential targets for PD interventions.

## 1. Introduction

Parkinson’s disease (PD) is an age-related neurodegenerative disease characterized by four clinical motor disorders: resting tremor, rigidity, bradykinesia, and postural instability [1]. Dopaminergic neuron loss in the substantia nigra pars compacta (SNpc) and inclusions in the cytoplasm (called Lewy bodies) are the major pathological changes in the brains of PD patients. The pathogenesis of PD is not fully understood. The treatment of PD is mainly symptomatic, and progressive neurodegenerative changes still occur over time.

Dopaminergic neurons in PD are particularly sensitive to oxidative stress. Oxidative stress is suggested to be one of the key factors in the pathogenesis of PD [2]. Oxidative damage has been detected in the brain tissues of both animal models of PD and PD patients [3]. The widely used PD animal models are induced by mitochondrial complex I inhibitors such as 6-hydroxydopamine (6-OHDA), 1-Methyl-4-phenyl-1,2,3,6-tetrahydropyridine (MPTP) and rotenone, which increase oxidative stress. A decrease in glutathione is one of the earliest detectable changes in the substantia nigra in patients with Parkinson’s disease (PD). Besides studies on oxidative stress in neurons per se, increasing evidence has shown that astrocytes play an important antioxidant role in the brain. Active astrocyte proliferation has been found around the areas of neuron loss both in the post-mortem tissue of PD patient brains and in animal models [4]. The proliferating astrocytes can release antioxidant molecules to protect dopaminergic neurons from oxidative damage [5]. However, the molecular mechanisms by which astrocytes protect against neurodegeneration are not clear.

The nuclear factor E2-related factor 2 (Nrf2)–antioxidant response element (ARE) pathway is one of the endogenous mechanisms counteracting the increased oxidative stress that occurs in the brains of PD patients. Nrf2 is a transcription factor controlled by Kelch ECH-associated protein 1 (Keap1), which targets the former for proteasomal degradation [6]. Under oxidative stress, the conformation of Keap1 changes, freeing Nrf2, which is stabilized and transferred from the cytoplasm to the nucleus, where it binds to the ARE to activate the expression of antioxidant enzymes, such as heme oxygenase-1 (HO-1) and glutamate–cysteine ligase catalytic (GCLC) subunit. Global Nrf2 overexpression and the use of Nrf2 activators have well-established neuroprotective effects in PD animal models [7,8], while Nrf2-knockout mice are more vulnerable to poisons such as 6-OHDA and rotenone, and the restoration of Nrf2 function in these animals can prevent the death of dopaminergic neurons. Our previous work has also demonstrated that 2-cyano-3,12-dioxooleana-1,9-dien-28-oic acid methyl ester (CDDO-Me, or called Bardoxolone Methyl) and selective overexpression of Nrf2 in Drosophila dopaminergic neurons delayed the progression of the neurodegeneration in a PD Drosophila model [9]. Studies have also shown that the overexpression of Nrf2 in astrocytes significantly protects against neurodegenerative diseases [10]. The overexpression of Nrf2 in neural stem cells can prevent neuronal damage induced by oxidative stress [11].

We previously found that autophagy flux was impaired by 6-OHDA treatment in SH-SY5Y cells and activation of autophagy displayed a neuroprotective effect in a 6-OHDA PD model [12]. It is suggested that the regulation of autophagy plays an important role in the pathogenesis of neurodegenerative disease [13]. To further understand the relationship between changes in the Nrf2–ARE signaling pathway and autophagy regulation in PD, we used both the 6-OHDA-induced rat PD model and rotenone-induced Drosophila PD model. We found that glial cells play an important role in the activation of the Nrf2–ARE signaling pathway, and Nrf2 activation is an upstream regulator of autophagy that protects against neurodegeneration in the PD models.

## 2. Materials and Methods

### 2.1. Animals

The Animal Ethics Committee of Soochow University (Suzhou, China) approved animal protocols on 2 March 2018. Healthy young adult Sprague-Dawley (SD) rats weighing 200–220 g were obtained from the SLAC Laboratory Animal Co., Ltd. (Shanghai, China). In order to maintain the consistency of research conditions, the male adult SD rats were used for experiments and kept in an animal house with 12 h light/12 h dark cycle under room temperature (25 °C) and humidity (60–80%) environment. All experiments were designed according to the guidelines of animal experimental center of Soochow University.

The flies were cultured on standard medium, at 25 °C and 60% relative humidity with a 12 h light/dark cycle. The 1 L standard medium consisted of 90 g Corn powder, 75 g sucrose, 36 g dry yeast, 10 g agar, 4.5 mL propionic acid, and 2 g nipagin. The *elav-GAL4* and *repo-GAL4* Drosophila lines were obtained from the Bloomington Stock Center (Indiana University, USA). *UAS-CncC* was obtained from Dirk Bohmann lab [14] and *UAS-CnccRNAi* fly line was from VDRC (Vienna Drosophila RNAi Center, v108127).

### 2.2. 6-OHDA Rat PD Model and CDDO-Me Treatment

A total of 48 rats were randomly divided into 4 groups (control group, 6-OHDA group, CDDO group, and 6-OHDA + CDDO group) after feeding one week in normal condition. The 6-OHDA group rats were treated with injections of 8 μL of 6-OHDA (2 μg/μL dissolved in 0.02% ascorbic acid saline solution) while the control group rats were injected with 8 μL saline. The injections were carried out using three-dimensional localization technique and were performed in the medial forebrain bundle (MFB, stereotaxic coordinates relative to Bregma were Anteroposterior: −2.2 mm, Lateral: −1.5 mm, Dorsoventral: −8.0 mm). In the CDDO group and 6-OHDA + CDDO group, CDDO-Me was given intraperitoneally with 0.5 mg/kg body weight for a continuous 3-day period after saline or 6-OHDA injection.

### 2.3. Apomorphine-Induced Rotational Experiment

The apomorphine-induced rotational experiments were performed one time per week in 5 weeks after the 6-OHDA injection. The apomorphine (0.5 mg/kg body weight) was subcutaneously injected in the back of the rats and the number of rotational behaviors to the contralateral side (opposite the lesioned side) was recorded after 5 min of apomorphine treatment for 30 min. Rats with a number of rotational behaviors to the contralateral side more than 7 turns/per min or 210 turns/30 min were considered as successful PD model rats.

### 2.4. Tissue Sampling and Immunohistochemistry

Rats were anesthetized with 4% chloral hydrate and perfused with 4% paraformaldehyde (PFA) in PBS. The brain sections were separated from whole brains by frozen section and subjected to immunofluorescent staining with mouse monoclonal anti-TH (1:1000, ImmunoStar, Hudson, WI, USA), rabbit monoclonal anti-Nrf2 monoclonal (1:100, Abcam, Cambridge, MA, USA), mouse monoclonal anti-NeuN (1:500, Millipore, Bedford, MA, USA), and mouse monoclonal anti-GFAP (1:300, Cell Signaling technology, Beverly, MA, USA) antibodies at 4 °C overnight. Then, the samples were incubated with the secondary antibody cy3 goat anti-mouse IgG (1:800, KPL, Gaithersburg, MD, USA) or 488 goat anti-rabbit IgG (1:400, KPL) for 1 h at room temperature. The nuclei were stained with Hoechst (1:100 dilute to final concentration of 1 µg/mL) at room temperature for 15 min. The images were captured immediately using a fluorescence microscope (AF6000, Leica, Wetzlar, Germany) and analyzed using ImageJ (National Institutes of Health, USA). Five brains were imaged separately for each experimental group.

### 2.5. Drosophila Model and Drug Treatment

For the Drosophila PD model, rotenone (Sigma-Aldrich, St. Louis, MO, USA) was added to the Drosophila food at final concentrations of 1, 10, and 100 µM. For drug treatment, the drugs were added to the fly food vials from the collection of newly eclosed flies. The Nrf2 inhibitor ML385 (S8790, Selleck, Houston, TX, USA), Nrf2 inducer CDDO-Me (S8078, Selleck), autophagy inhibitor 3-MA (S2767, Selleck), and autophagy inducer rapamycin (S1039, Selleck) were added to the Drosophila food at a final concentration of 10 µM. All the food vials were changed every other day. For overexpression and knockdown of drosophila Nrf2 orthologue “cncc”, we used the hybridization of UAS-cncc or UAS-cnccRNAi fly lines with elav-Gal4 (expression of target genes in neurons) and Repo-GLA4 (expression of target genes in glial cells).

### 2.6. The Drosophila Lifespan Test

Next, 20–25 newly eclosed female flies were collected in a food vial and 4 parallel vials were performed for each experiments group. The food vials were changed every other day, and the numbers of surviving flies in each vial were counted. The survival ratio was calculated to create the survival curve. The Kaplan–Meier survival curves with Log-rank test was used to analyze survival curves by the GraphPad Prism software (GraphPad Software Inc., La Jolla, CA, USA), and the half survival time (the time for 50% fly survival in each vials) was analyzed for a special single point of different groups using one-way ANOVA; *p* < 0.05 was considered as statistically significant difference.

### 2.7. Climbing Ability Test

The climbing ability test was performed as previously described with minor modifications [9]. Each experimental group of 25 fruit flies was placed in an empty culture tube (15 cm, 1.5 cm in diameter). After 30 min of rest at room temperature, all the flies were flicked to position them on the bottom of the tube. The numbers of Drosophila climbing to or above the test line (10 cm above the bottom) within 10 s were counted, and the data were analyzed based on three repeats of the assay each week.

### 2.8. Redox State Analysis of the Drosophila Brains

The fluorescent dye dihydroethidium (DHE, D11347, Invitrogen, San Diego, CA, USA) was used to monitor the redox state in the brains of the Drosophila. The DHE was dissolved in DMSO (at a 30 mM concentration) to create a stock solution and diluted in Schneider’s medium (to a 30 µM final concentration) for the working solution. The brains of the Drosophila were dissected in Schneider’s medium, incubated in 30 µM DHE in a dark chamber for 10 min at room temperature, rinsed 3 times with PBS in a dark chamber, and mounted in mounting medium (H-1000, Vector, Burlingame, CA, United States). The images were captured immediately using a fluorescence microscope (AF6000, Leica, Wetzlar, Germany) and analyzed using ImageJ. Five brains were imaged separately for each experimental group.

### 2.9. Tissue Preparation and Western Blotting

For the rats, the brains were dissected to separate the substantia nigra and striatum for homogenization in RIPA Lysis Buffer 10 μL/per mg tissues (Beyotime Biotechnology, Shanghai, China, 50 mM Tris-HCl, pH7.4, 150 mM NaCl, 1% Triton X-100, 1% sodium deoxycholate, 0.1% SDS) with protease inhibitor PMSF (Beyotime, RIPA: PMSF = 100:1). For Drosophila, 5 heads of each experimental group were homogenized in RIPA Lysis Buffer with protease inhibitor. After 12.5% SDS–PAGE gel (Epizyme Biotech, Shanghai, China) electrophoresis, the proteins were transferred onto a PVDF membrane (Millipore). The membrane was then blocked with 5% nonfat milk in TBST and then incubated with primary antibodies overnight at 4 °C and, subsequently, with secondary antibody for 2h at room temperature. The primary antibodies used for the Western blot included anti-Nrf2 (1:1000, Assaybiotech), anti-GCLC (1:1000, BD), anti-HO-1 and anti-LC3I/II (Sigma), anti-Ref(2)P (1:1000, BD Pharmingen, San Jose, CA, USA), and anti-β-tubulin (1:5000, Sigma-Aldrich). The bands were detected by using chemiluminescence substrates (Beyotime Biotechnology) and analyzed with the NIH ImageJ software. The experiments of WB were repeated at least three times and we performed the statistical analysis of the results using GraphPad Prism (GraphPad Software Inc., La Jolla, CA, USA).

### 2.10. Quantitative Real Time Polymerase Chain Reaction (qRT-PCR)

Total RNA was extracted from the 5 heads of drosophila using Trizol reagent (Invitrogen) according to the protocol. cDNA was synthesized using the Reverse Transcription System Kit (Takara, Beijing, China). The qRT-PCR test was performed by SYBR Green PCR Master Mix (Takara) and detected on ABI 7500 system (Life technology, Carlsbad, CA, USA). The primers for cncc were 5′-GAGGTGGAAATCGGAGATGA-3′ and 5′-CTGCTTGTAGAGCACCTCAGC-3′, the β-actin was used to normalize the relative level of gene expression and the primers were 5′- GGA GAT TAC TGC CCT GGC TCC TA-3′ and 5′-GGA GAT TACTGC CCT GGC TCC TA-3′.

### 2.11. Statistical Analysis

The data are presented as the mean ± standard error of the mean and were analyzed by one-way ANOVA, and the survival curves were analyzed by Log-rank test using the GraphPad Prism software. Differences were considered statistically significant at *p* < 0.05.

## 3. Results

### 3.1. CDDO-Me Significantly Attenuated 6-OHDA-Induced PD-Like Phenotypes in Rats

The 6-OHDA was unilaterally injected into the MFBs of rats to induce typical PD-like behavioral and pathological changes. CDDO-Me was administered intraperitoneally at 0.5 mg/kg for 3 days continuously after 6-OHDA injection in the 6-OHDA + CDDO group. We executed an apomorphine-induced contralateral rotation experiment to test the behavioral change. Our results show that the control group (sham operation group) showed no contralateral rotation behavior after apomorphine treatment. In the 6-OHDA group, the contralateral rotation behavior induced by apomorphine appeared in 1 week after 6-OHDA injection, and the number of rotations increased significantly after 2 weeks and reached a plateau at 4 to 5 weeks after 6-OHDA injection. After 6 weeks, the number of rotations slowly decreased (Figure 1A). Statistical analysis showed that the apomorphine-induced rotation significantly increased in the 6-OHDA group compared with the control group throughout the test period, indicating that 6-OHDA injection causes this typical PD-like symptom.

Treatment with CDDO-Me (an Nrf2 inducer) decreased the rotation number in the 6-OHDA + CDDO group, according to comparison with in the 6-OHDA group, and statistical analysis showed a significant difference between the 6-OHDA + CDDO group and 6-OHDA group at the third week after CDDO-Me treatment (Figure 1A). The survival of dopaminergic neurons was assessed by tyrosine hydroxylase (TH) immunostaining. The number of TH-positive neurons in the SNpc was significantly lower in the 6-OHDA group than in the control group at 2 weeks after the 6-OHDA injection (Figure 1B). In the 6-OHDA + CDDO group, there were around twice as many TH-positive neurons in the SNpc compared to the 6-OHDA group (Figure 1B). The proliferation of astrocytes was measured by GFAP immunostaining. Our results show a significantly increased number of active astrocytes in the SNpc of the 6-OHDA group compared with that in the control group (Figure 1C). CDDO-Me treatment decreased the proliferation of active astrocytes compared with that in the 6-OHDA exposure group (Figure 1C).

### 3.2. CDDO-Me Induced Activation of Nrf2 Pathway in Astrocytes and Increased Autophagy in 6-OHDA Rat Model

We found that the Nrf2 fluorescence was mainly distributed in the cytoplasm of neurons in the brain slices of the control group. Injection with 6-OHDA significantly increased the number of Nrf2-positive cells (Figure 2A). Moreover, 6-OHDA also induced the translocation of Nrf2 to the nucleus (Figure 2A). CDDO-Me treatment further increased the number of Nrf2-positive cells and nuclear Nrf2 staining (indicating nuclear translocation), as seen by comparing the 6-OHDA + CDDO group with the 6-OHDA group (Figure 2A). The co-staining of Nrf2 with GFAP showed that there was more than twice the number of stain-positive astrocytes in the 6-OHDA group than the vehicle group and a further increase in the 6-OHDA + CDDO-Me group (Figure 2B), indicating that CDDO-Me further activated the Nrf2 signaling pathway in astrocytes.

The expression of the Nrf2 protein and its downstream antioxidant protein GCLC was also assessed by Western blotting of homogenates of the substantia nigra and striatum from rat brains in each group at 2 weeks after 6-OHDA injection. The protein levels of Nrf2 and GCLC increased in both the substantia nigra and striatum in the 6-OHDA group compared with those in the control group. These were further increased in the 6-OHDA + CDDO group, indicating that CDDO-Me effectively activated the Nrf2 signaling pathway in vivo in the PD rat model (Figure 2C).

Previous studies showed that 6-OHDA leads to the impairment of autophagic inward flow in cultured neuroma cells [12]. To determine whether CDDO-Me altered the autophagy pathway, we measured the levels of the autophagy-associated protein LC3-II, to assess the level of autophagy in the brain tissue of rats in each experimental group. The results show that LC3-II was not altered in the 6-OHDA group, but CDDO-Me significantly increased the LC3-II levels in both the substantia nigra and striatum of the 6-OHDA + CDDO group compared with those of the 6-OHDA group (Figure 2D).

### 3.3. CDDO-Me Significantly Improved the Flies’ Lifespans and Motility in a PD Drosophila Model

Like 6-OHDA, rotenone induces oxidative damage and dopaminergic neuron loss and leads to a PD-like change in Drosophila [15]. To further assess and validate the protective effect of CDDO-Me, a rotenone-based PD Drosophila model was used. Rotenone exposure at 1, 10, and 100 μmol/L in fly food dose-dependently shortened the survival time. CDDO-Me at 10 μmol/L in fly food significantly reduced the rotenone-induced shortening of lifespan (Figure 3A). The Log-rank test and half survival time was used for the statistical analysis of lifespan among the different groups. Rotenone significantly reduced the survival time in a dose-dependent manner (Figure 3B). CDDO-Me treatment increased the survival time in both the vehicle and rotenone exposure groups (Figure 3B). The motility assay in Drosophila also showed a protective effect of CDDO-Me administration (Figure 3C). At 3–4 weeks of age, rotenone significantly reduced the flies’ locomotor ability. CDDO-Me treatment attenuated the rotenone-induced locomotor impairment (Figure 3C).

### 3.4. Overexpression of Nrf2 in Glial Cells Displayed Antioxidant Effects and Rescued Rotenone-Induced PD-Like Phenotypes in Flies

The data from the PD rat experiments showed that the activation of the Nrf2 signaling pathway was more pronounced in glial cells in the brain. To elucidate whether Nrf2 activation in glial cells acted differently than that in neurons, we used Drosophila as a model to selectively express Nrf2 in neurons and glial cells. We obtained elav-Gal4;UAS-cncc (Elav;cncc) through Drosophila hybridization to overexpress the Nrf2 Drosophila orthologue “cncc” in neurons, and Repo-GLA4;UAS-cncc (Repo;cncc) to overexpress Nrf2 in glial cells. The elav-Gal4;UAS-cncc (Elav;cncc) and Repo-GLA4;UAS-cncc flies were left alone or exposed to 10 μmol/L rotenone for all the experimental days. The expression of Nrf2 in both neurons and glial cells effectively prolonged the survival time of rotenone-based PD flies (Figure 4A). Rotenone caused survival times shorter than those in the DMSO control. The overexpression of Nrf2 in both neurons and glial cells prolonged the survival time in the rotenone/Elav;cncc group and rotenone/Repo;cncc group compared to that for flies only exposed to rotenone in Rotenone/WT group (Figure 4A). The overexpression of Nrf2 in the glial cells in the rotenone/Repo;cncc group improved the survival more than Nrf2 expression in neurons in the rotenone/Elav;cncc group (Figure 4A).

Nrf2 overexpression in both neurons and glial cells significantly ameliorated the rotenone-induced locomotor impairment at 3–5 weeks of age (Figure 5B). Nrf2 overexpression in glial cells in the rotenone/Repo;cncc group showed a better protective effect than Nrf2 expression in neurons in the rotenone/Elav;cncc group (Figure 4B). The protein expression levels of the Nrf2 downstream antioxidant GCLC and HO-1 increased more significantly in the rotenone/Repo;cncc group than in the rotenone/Elav;cncc group (Figure 4C). The redox staining with DHE showed an increased level of oxidative stress after rotenone exposure in flies. Nrf2 overexpression attenuated rotenone-induced oxidative stress, and Nrf2 overexpressing in glial cells in the rotenone/Repo;cncc group had a lower level of DHE staining than Nrf2 overexpression in neurons in the rotenone/Elav;cncc group (Figure 4D).

### 3.5. Activation of the Nrf2 Signaling Pathway Enhanced Autophagy

In the PD model rats, we found that CDDO-Me significantly increased the autophagic flux. To further assess the effects of Nrf2 overexpression on the autophagic flux in the Drosophila PD model, we measured the levels of the Ref(2)P protein, which is a marker of autophagy flux in drosophila. Ref(2)P did not change in the rotenone-exposed flies compared with DMSO control flies. Nrf2 overexpression in neurons in rotenone/Elav;cncc flies or glial cells in rotenone/Repo;cncc flies increased Ref(2)P. Nrf2 overexpression in glial cells led to higher Ref(2)P protein levels than in neurons (Figure 4C), suggesting that the activation of the Nrf2 signaling pathway in glial cells has a stronger ability to enhance autophagy, which may contribute to the protective effects against rotenone-induced neurotoxicity.

To further assess whether the Nrf2-linked antioxidant signaling pathway alters autophagy, we treated rotenone/Repo;cncc flies with the Nrf2 inhibitor ML385, Nrf2 inducer CDDO-Me, autophagy inhibitor 3-MA, and autophagy inducer rapamycin. The overexpression of Nrf2 in glial cells in the Rot/Repo;cncc group prolonged the survival time (Figure 5A), increased the expression of the Nrf2 downstream antioxidant protein HO-1 and increased the level of Ref(2)P (Figure 5B). ML385 shortened the survival time of rotenone/Repo;cncc flies, suppressed the neuroprotective effect of Nrf2 (Figure 5A), and reduced HO-1 and Ref(2)P levels, according to comparison with the DMSO control group (Figure 5B).

The autophagy inhibitor 3-MA also shortened the survival time of rotenone/Repo;cncc flies and attenuated the neuroprotective effect of Nrf2 overexpression by inhibiting autophagy (Figure 5A,B). However, 3-MA did not alter the HO-1 protein level (Figure 5B). Treatment with CDDO-Me or the autophagy inducer rapamycin did not further increase the survival time or HO-1 and Ref(2)P protein levels in rotenone/Repo;cncc flies (Figure 5B). It is possible that the overexpression of Nrf2 maximally activated the autophagy system, so CDDO-Me and rapamycin could not further activate it.

### 3.6. Knockdown of Nrf2 in Fly Glial Cells Worsened Rotenone-Induced PD-Like Phenotypes in Flies

To further assess Nrf2-linked pathways, we used UAS-cnccRNAi to knockdown Nrf2 in glial cells of the rotenone-based PD Drosophila model. For the detection of Nrf2 orthologue cncc level in Repo-GLA4; UAS-cnccRNAi flies, the qRT-PCR were performed by primers 5′-GAGGTGGAAATCGGAGATGA-3′ and 5′-CTGCTTGTAGAGCACCTCAGC-3′. The cncc level in Repo;cnccRNAi flies was about 40% compared with Control (Figure 6B). The knockdown of Nrf2 worsened rotenone-induced toxicity in Rot/Repo;cnccRNAi flies by further shortening the survival time (Figure 6A) and decreasing the levels of HO-1 and autophagy proteins compared with those in the rotenone/WT group (Figure 6C). Treatment with CDDO-Me almost abrogated the Nrf2-knockout effects in Rot/Repo;cnccRNAi flies (Figure 6A,C). Moreover, the autophagy inducer rapamycin also partially reversed the effects of Nrf2 knockout on the survival time and Ref(2)P protein expression in Rot/Repo;cnccRNAi flies, but it did not alter HO-1 levels (Figure 6C). By contrast, the autophagy inhibitor 3-MA further reduced the expression of the autophagy protein Ref(2)P in Nrf2-knockdown Drosophila, but did not alter the expression of HO-1 (Figure 6C).

## 4. Discussion

The main findings of this study are that increased Nrf2 expression in glial cells (astrocytes) protected against 6-OHDA-inudced PD-like phenotypes in rats and rotenone-induced PD-like phenotypes in flies via upregulating the endogenous antioxidant pathway and enhancing autophagy. In addition, treatment with CDDO-Me (an Nrf2 inducer) increased the antioxidant signaling pathway and enhanced autophagy in the substantia nigra and striatum of the PD rats. In PD flies, treatment with autophagy inducer rapamycin enhanced the autophagy pathway, but did not alter the antioxidant pathway, thereby only partially rescued the PD-like phenotypes. Meanwhile, treatment with the Nrf2 inhibitor ML385 or autophagy inhibitor 3-MA significantly worsened the rotenone-induced PD-like phenotypes in flies. Moreover, the knockdown of Nrf2 in glial cells in flies worsened the rotenone-induced PD-like phenotypes, further validating that the Nrf2 antioxidant pathway and autophagy pathway play important roles in PD pathogenesis.

Oxidative-stress-induced brain damage is an important mechanism of injury in neurodegenerative diseases including PD. Recent studies suggest that the Nrf2–ARE antioxidant signaling pathway plays an important role in protecting against oxidative-stress-induced injury [16]. The 6-OHDA-based PD rat model [17] or rotenone-based fly model [18] is often used to study oxidative-stress-linked pathways and potential treatment strategies for PD. In this study, we found that activation of Nrf2 signaling pathway in astrocytes plays a very important neuroprotective role in the animal model of PD. Astrocytes play contradictory roles in PD pathogenesis. On one hand, astrocytes are considered “good” glia due to their capacity to remove toxic molecules and for releasing neurotrophic factors such as ascorbic acid salt, superoxide dismutase (SOD), and glutathione [19,20,21]. Similar to our findings, transplantation of astrocytes into PD mice can activate Nrf2 and counteract motor function defects [22]. However, under pathological conditions, astrocytes can also facilitate neurodegeneration by releasing toxic molecules [23,24]. Harnessing this double-bladed sword may be pivotal for resolving the mystery of PD [25] and discovering of useful therapeutic targets [26,27].

The study of Nrf2 has recently been popular among researchers, because the Nrf2–ARE antioxidant signaling pathway plays a very important role in the pathogenesis of neurodegenerative diseases. The expression of antioxidant proteins by Nrf2 activation produces important defensive effect against oxidative damage in neurons [9]. In this study, we found that the activation of the Nrf2 signaling pathway through CDDO-Me treatment protected against 6-OHDA-induced PD-like phenotypes in rats. CDDO-Me is a compound derived from the terpenoidoleanolic acid and can react with and change the conformation of Keap1to induce the expression of downstream antioxidant genes [28]. CDDO-Me ameliorated the 6-OHDA-induced locomotor impairment and loss of TH-positive neurons. Meanwhile, CDDO-Me effectively activated the Nrf2/ARE signaling pathway and autophagy in the substantia nigra and striatum in the 6-OHDA + CDDO-Me rats. Nrf2 regulates the expression of a series of cellular protective proteins involved in ROS clearance, the xenobiotic metabolism, and detoxifying functions [29,30]. The activation of the Nrf2 signaling pathway can activate the downstream antioxidant genes GCLC and HO-1 to protect against nerve damage caused by 6-OHDA in cells and rats [31,32]. Interestingly, in SNpc slices of rat brains, we found that Nrf2-positive immunoreactivity was mainly located in the cytoplasm in the normal control group, and 6-OHDA induced Nrf2 redistribution and translocation to the nucleus in rat brain cells. Most importantly, the co-staining of Nrf2 with GFAP showed that Nrf2 was mainly expressed in the cytoplasm of neurons and less in astrocytes in the control group, but Nrf2-positive signals were mainly found in the nuclei of astrocytes in 6-OHDA-treated rats. Moreover, the CDDO-Me-induced Nrf2 expression was also predominantly found in GFAP-positive astrocytes in the 6-OHDA + CDDO-Me rats. This suggests that Nrf2 in astrocytes may play an important role in protecting against 6-OHDA-induced oxidative damage. While CDDO-Me activates the Nrf2 pathway in astrocytes, it also induces the enhancement of an autophagy signal in the PD rat model.

The impairment of autophagy flux is suggested to be another contributing factor for PD-like phenotypes [33]. Autophagy is a conserved process in which cells degrade proteins or impaired organelles to maintain cellular homeostasis [34]. The impaired autophagy was observed in various regions of the brain in PD patients [35]. To further understand the neuroprotective role of the Nrf2-induced antioxidant pathway and autophagy pathway, we used the rotenone-based PD fly model, as the flies have short lifespans and there are various genetic and pharmacological tools with which to up- and downregulate these pathways [36]. Our results showed that rotenone reduced the survival time and caused motor dysfunction in flies, consistent with our results from the 6-OHDA rat model. And the CDDO-Me administration rescued rotenone-induced PD-like phenotypes, including improving the survival time and motor ability. The overexpression of the Nrf2 orthologue “cncc” in glial cells (Repo-gla4;UAS-cncc) in rotenone-induced PD flies displayed more protective effects than that in neurons (Elav-Gal4;UAS-cncc). These neuroprotective effects were probably due to decreased oxidative stress by the activation of an antioxidant signaling pathway as well as enhanced autophagy levels in the PD flies. The studies demonstrated that the mitochondrial derived ROS production increases autophagy [37] and enhanced autophagy can also be targeted to impaired mitochondria (mitophagy) [38]. In order to further elucidate the relationship between antioxidant signaling pathway and autophagy, we conducted administration experiments of Nrf2 inhibitor ML385, autophagy inhibitor 3-MA, Nrf2 inducer CDDO-Me, and autophagy inducer Rapamycin in PD model drosophila with overexpressed or knockdown Nrf2 in glia cells. We observed that treatment with the Nrf2 inducer CDDO-Me or autophagy inducer rapamycin suppressed rotenone-induced PD-like phenotypes. By contrast, treatment with the Nrf2 inhibitor ML385 or autophagy inhibitor 3-MA worsened rotenone-induced PD-like phenotypes in flies. Our results indicate that autophagy plays a role downstream of the Nrf2 signaling pathway; activation of the Nrf2 signaling pathway can effectively enhance autophagy protein levels, and activation of autophagy cannot significantly affect antioxidant proteins in the Nrf2 signaling pathway. Taken together, these results demonstrate that Nrf2 upregulation protected against 6-OHDA- or rotenone-induced PD-like phenotypes due to increased autophagy as well as increased the activity of the antioxidant defense system. We also think that mitophagy was involved in our PD models and enhanced mitophagy may improve oxidative stress to protect from neurodegeneration. We need to do more experiments to test this in the future.

In summary, increased Nrf2 expression in glial cells (astrocytes) protected against 6-OHDA-induced PD-like phenotypes in rats and rotenone-induced PD-like phenotypes in flies via upregulating the antioxidant pathway and enhancing autophagy. Our study not only provides a novel insight into the Nrf2-linked signaling mechanisms underlying PD, but also offers potential targets for PD intervention.

## Figures and Tables

**Figure 1 cells-10-01850-f001:**
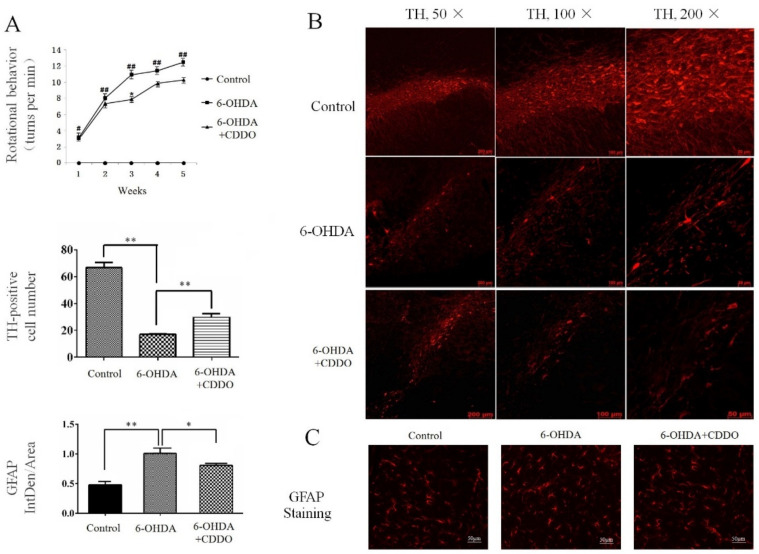
CDDO-Me improved 6-OHDA-induced PD-like phenotypes in rats. (**A**) Apomorphine-induced contralateral rotation assays. The number of contralateral rotations significantly increased in 6-OHDA rats compared with that in the control group (^#^
*p* < 0.05, ^##^
*p* < 0.01, *n* = 5, compared with the control group). The rotation behavior appeared in first week after 6-OHDA injection (average rotation score was 3.13 ± 0.62 r/min, ^#^
*p* < 0.05, *n* = 5, compared with the control group), increased in 2–5 weeks after 6-OHDA injection (average rotation score was 8.07 ± 0.79 r/min at week 2, 10.95 ± 0.94 r/min at week 3, 11.42 ± 0.81 r/min at week 4, ^##^
*p* < 0.01, *n* = 5), and reached a plateau at week 5 (average rotation score was 12.53 ± 0.84 r/min, ^##^
*p* < 0.01, *n* = 5). The rotation number began to fall at week 6 after 6-OHDA injection. CDDO-Me decreased rotation performance in the 6-OHDA + CDDO group compared with that in the 6-OHDA group during all test weeks, and a significant difference was observed at week 3 after CDDO-Me treatment (average rotation score was 7.86 ± 0.73 r/min, * *p* < 0.05, *n* = 5, compared with the 6-OHDA group). (**B**) TH immunostaining. Representative pictures show TH-positive dopaminergic neurons in SNpc at week 2 after 6-OHDA injection (50×, 100× and 200× magnification). The bar graph on the left side of the pictures is the quantitative analysis of the number of TH-positive dopaminergic neurons. The results show dopaminergic neurons were significantly reduced in the 6-OHDA group compared with in the control group (** *p* < 0.01, *n* = 5). CDDO-Me significantly increased the number of dopaminergic neurons in the 6-OHDA + CDDO group compared with in the 6-OHDA group (** *p* < 0.01, *n* = 5). (**C**) GFAP immunostaining. Representative pictures show GFAP-positive astrocytes in SNpc at Week 2 after 6-OHDA injection. The bar graph on the left side of the pictures is the quantitative analysis of optical density for GFAP fluorescence. The fluorescence of GFAP-positive astrocytes significantly increased in the 6-OHDA group compared with in the control group (** *p* < 0.01, *n* = 5). CDDO-Me reduced fluorescence density of reactive astrocytes in the 6-OHDA + CDDO group compared with in the 6-OHDA group (* *p* < 0.05, *n* = 5).

**Figure 2 cells-10-01850-f002:**
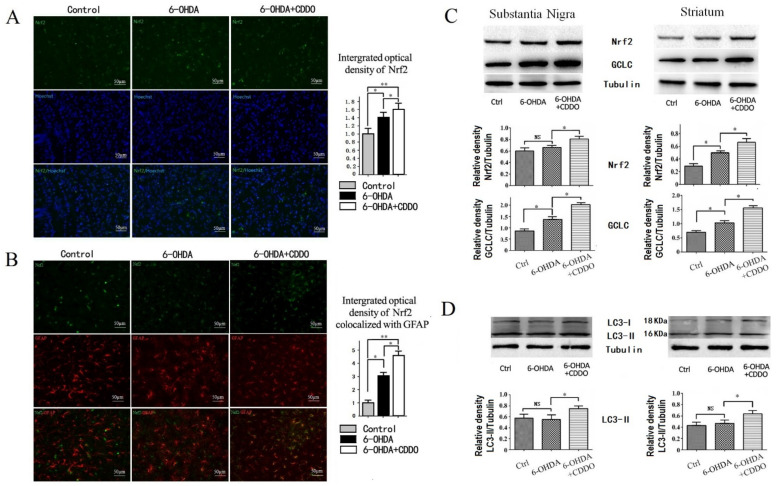
CDDO-Me activated Nrf2/ARE pathway predominantly in astrocytes in 6-OHDA rat brains. (**A**) Nrf2 immunostaining. Representative pictures show Nrf2 immunofluorescence in SNpc at week 2 after 6-OHDA injection, and nuclei were re-stained with Hoechst (200× magnification). The bar graph on the right side of the pictures is the quantitative analysis of optical density for fluorescence. The Nrf2-positive cells were few in the control group and mainly distributed in the cytoplasm. The Nrf2 signal increased in the 6-OHDA group compared with the control group (* *p* < 0.05, *n* = 5) and was mainly distributed in the nucleus. CDDO-Me increased Nrf2-positive cells in the 6-OHDA + CDDO group compared with the 6-OHDA group and control group (* *p* < 0.05,** *p* < 0.01, *n* = 5), and the Nrf2 signal was also mainly distributed in the nucleus in the 6-OHDA + CDDO group. (**B**) Co-immunostaining of Nrf2 with GFAP. Representative pictures show Nrf2 fluorescence was mainly distributed in neurons but not in GFAP-positive astrocytes in control group. Increased Nrf2 signal was colocalized with GFAP staining in the 6-OHDA group and 6-OHDA + CDDO group. The bar graph on the right side of the pictures is the quantitative analysis of optical density for Nrf2 fluorescence colocalization with GFAP-positive astrocytes. The results show more Nrf2 signal colocalized with GFAP fluorescence in the 6-OHDA group compared with the control group (* *p* < 0.05, *n* = 5). CDDO-Me further increased the Nrf2-positive signal in GFAP-positive astrocytes in the 6-OHDA + CDDO group compared with the 6-OHDA group (* *p* < 0.05, ** *p* < 0.01, *n* = 5). (**C**) Representative Western blots of Nrf2 and GCLC proteins in the substantia nigra and striatum homogenate. The bar graph below is the quantitative analysis of Western blot bands. Nrf2 did not significantly change in the substantia nigra (NS *p* > 0.05) and increased in the striatum in the 6-OHDA group compared with the control group (* *p* < 0.05). CDDO-Me increased Nrf2 in both substantia nigra and striatum in the 6-OHDA + CDDO group compared with in the 6-OHDA group (* *p* < 0.05). The Nrf2 downstream antioxidant GCLC also significantly increased in the substantia nigra and striatum in the 6-OHDA group compared with the control group (* *p* < 0.05). CDDO-Me further increased the GCLC level in the 6-OHDA + CDDO group compared with in the 6-OHDA group (* *p* < 0.05). (**D**) Representative Western blots of LC3 and quantitative analysis. The expression of the autophagy-related protein LC3-II did not significantly change in the substantia nigra and striatum in the 6-OHDA group compared with the control group (NS *p* > 0.05). CDDO-Me effectively increased LC3-II expression in the 6-OHDA + CDDO group compared with the 6-OHDA group (* *p* < 0.05).

**Figure 3 cells-10-01850-f003:**
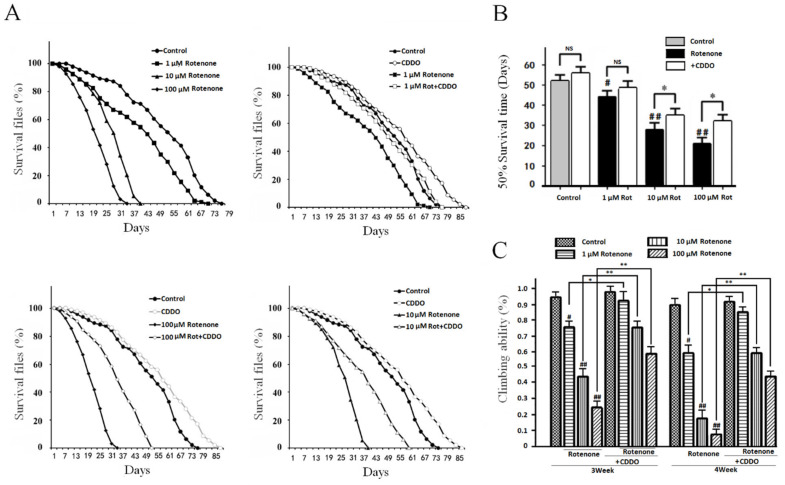
CDDO-Me treatment improved the survival time and motor ability in rotenone PD Drosophila model. (**A**) The fly survival curve. All three concentrations of Rotenone reduced survival time compared with the control group (*p* < 0.05, 1, 10, and 100 µmol/L Rotenone vs. DMSO control flies; the survival curves were analyzed by Log-rank test). The 10 µmol/L CDDO-Me administration extended survival time in all three concentration of Rotenone (*p* < 0.05 by Log-rank test, CDDO vs. 1, 10, and 100 µmol/L Rotenone). (**B**) Half survival time. Rotenone reduced half survival time. (^#^
*p* < 0.05 and ^##^
*p* < 0.01 vs. DMSO control flies). CDDO-Me significantly increased half survival time in 10 and 100 µmol/L rotenone groups (* *p* < 0.05), but failed to extend half survival time in the control and 1 µmol/L rotenone groups (NS *p* > 0.05). (**C**) Climbing ability assays. The climbing ability was assayed weekly, and the bar graph shows the results at the age of 3–4 weeks. Doses of 1, 10, and 100 µmol/L rotenone significantly reduced climbing ability compared with the control group (^#^
*p* < 0.05 and ^##^
*p* < 0.01), and CDDO-Me administration improved rotenone-induced motor dysfunction compared with the corresponding rotenone group (* *p* < 0.05 and ** *p* < 0.01).

**Figure 4 cells-10-01850-f004:**
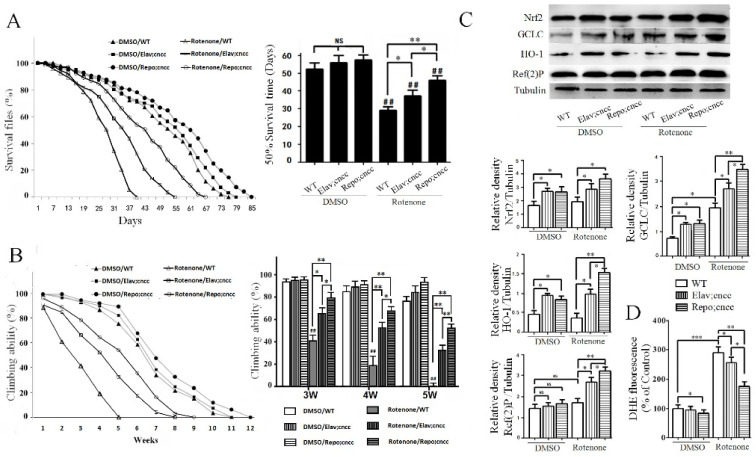
Overexpression of Nrf2 in glial cells suppressed rotenone-induced PD-like phenotypes in flies. (**A**) survival curve and statistical analysis by Log-rank test and half survival time. Nrf2 overexpression in DMSO control drosophila did not significantly improve survival time in the DMSO/Elav;cncc or DMSO/Repo;cncc group compared with the DMSO/WT control group (NS *p* > 0.05). Rotenone treatment significantly shortened survival time compared with the corresponding DMSO control group (^##^
*p* < 0.01). Nrf2 overexpression in neurons of PD model Drosophila significantly increased survival time in the rotenone/Elav;cncc group compared with the rotenone/WT group (* *p* < 0.05). Nrf2 overexpression in glial cells more significantly increased survival time in the rotenone/Repo;cncc group compared with the rotenone/Elav;cncc group and the rotenone/WT group (* *p* < 0.05 and ** *p* < 0.01). (**B**) The left graph shows climbing ability throughout lifetime. The right bar graph shows the climbing ability at ages of 3–5 weeks. Rotenone treatment significantly reduced climbing ability in the rotenone/WT group compared with the DMSO group (^##^
*p* < 0.01). Nrf2 overexpression in glial cells in the rotenone/Repo;cncc group more significantly improved the motor ability than overexpression in neurons in the rotenone/Elav;cncc group (* *p* < 0.05 and ** *p* < 0.01). (**C**) Western blot analysis of Nrf2, GCLC, HO-1, and Ref(2)P proteins in Drosophila brain homogenates. The bar graph below is the quantitative analysis of Western blot bands. Nrf2 overexpression in neurons and glial cells increased GCLC and HO-1 in the Elav;cncc and Repo;cncc group compared with the WT control group (* *p* < 0.05 and ** *p* < 0.01). Nrf2 overexpression in glial cells further increased Nrf2, GCLC and HO-1 in the Repo;cncc group compared to overexpression in neurons in the Elav;cncc group (* *p* < 0.05). The Nrf2 overexpression significantly increased Ref(2)P in the rotenone/Elav;cncc and rotenone/Repo;cncc groups compared with the rotenone/WT group (* *p* < 0.05 and ** *p* < 0.01), but not in the DMSO control group. Nrf2 overexpression in glial cells enhanced autophagy more than in neurons (* *p* < 0.05). (**D**) Measure of redox state by DHE staining. Oxidative stress was relatively low in the DMSO control group. Rotenone treatment significantly increased oxidative stress in the rotenone/WT group compared with the DMSO control group (*** *p* < 0.001). Nrf2 overexpression slightly reduced oxidative stress in the DMSO control group and significantly decreased rotenone-induced oxidative stress in the rotenone/WT group (* *p* < 0.05 and ** *p* < 0.01). Nrf2 overexpression in glial cells in the rotenone/Repo;cncc group resulted in more DHE staining than such overexpression in neurons in the rotenone/Elav;cncc group (* *p* < 0.05).

**Figure 5 cells-10-01850-f005:**
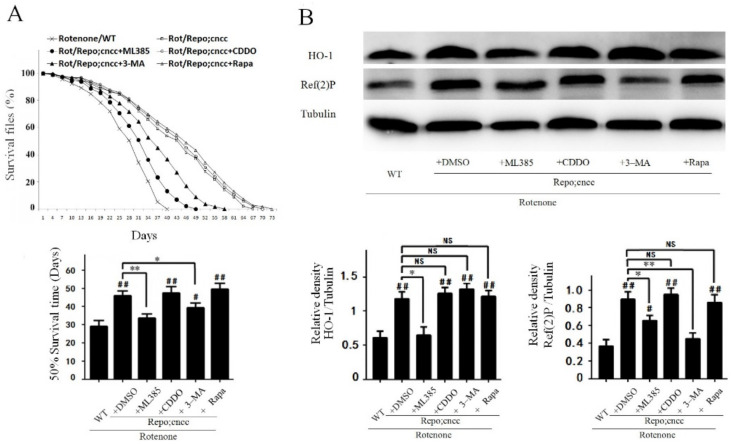
Activation of the Nrf2 signaling pathway enhanced autophagy in flies. (**A**) The graph shows the lifetime survival curve, the Log-rank test, and half survival time used to analyze the difference between groups (^#^, compared with the rotenone/WT group). Nrf2 overexpression in glial cells effectively prolonged survival time in the Rot/Repo;cncc + DMSO group compared with the rotenone/WT group (^##^
*p* < 0.01). The Nrf2 inhibitor ML385 significantly reduced the prolonged survival time in the Rot/Repo;cncc + ML385 group compared with the Rot/Repo;cncc + DMSO group (** *p* < 0.01). The autophagy inhibitor 3-MA partly reduced the prolonged survival time of the Rot/Repo;cncc + 3-MA group (* *p* < 0.05). The Nrf2 inducer CDDO-Me and autophagy inducer rapamycin did not significantly affect the survival time of the Rot/Repo;cncc + DMSO group. (**B**) Western blot analysis of antioxidant and autophagy proteins in fly brain. Nrf2 overexpression in glial cells significantly increased HO-1 and Ref(2)P proteins in the rotenone/Repo;cncc group compared with the rotenone/WT group (^##^
*p* < 0.01). The Nrf2 inhibitor ML385 effectively decreased HO-1 and Ref(2)P levels in the rotenone/Repo;cncc + ML385 group compared with the rotenone/Repo;cncc + DMSO group (* *p* < 0.05). The autophagy inhibitor 3-MA effectively decreased Ref(2)P protein but not HO-1 protein, while the Nrf2 inducer CDDO-Me and autophagy inducer rapamycin did not significantly affect the expression of HO-1 and Ref(2)P in the rotenone/Repo;cncc + DMSO group.

**Figure 6 cells-10-01850-f006:**
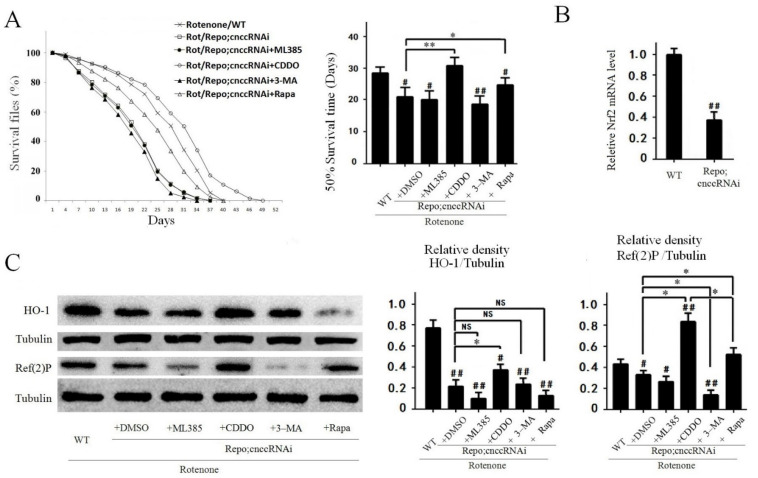
Knockdown of Nrf2 in fly glial cells worsens rotenone-induced PD-like phenotypes in flies. (**A**) The survival curve in Nrf2-knockdown Drosophila. The graph shows the survival curve, the Log-rank test, and half survival time used to analyze the difference between groups (^#^, compared with the rotenone/WT group). Nrf2 knockdown in glial cells reduced survival time in the Rot/Repo;cnccRNAi + DMSO group compared with the rotenone/WT group (^#^
*p* < 0.05). CDDO-Me significantly extended survival time in the Rot/Repo;cnccRNAi + CDDO-Me group compared with the Rot/Repo;cnccRNAi + DMSO group (** *p* < 0.01). Rapamycin partially restored the shortened survival time of the Rot/Repo;cnccRNAi + DMSO group (* *p* < 0.05). (**B**) qRT-PCR result for drosophila Nrf2, the cncc level in Repo;cnccRNAi flies was about 40% compared with WT Control (^##^
*p* < 0.01). (**C**) Western blot analysis of antioxidant and autophagy proteins in fly brain. Nrf2 knockdown in glial cells by RNAi significantly decreased protein expression of HO-1 and Ref(2)P in the rotenone/Repo;cnccRNAi group compared with the rotenone/WT group (^#^
*p* < 0.05 and ^##^
*p* < 0.01). CDDO-Me partially restored this decrease in HO-1 and Ref(2)P expression in the rotenone/Repo;cnccRNAi + CDDO-Me group compared with the rotenone/Repo;cnccRNAi + DMSO group (* *p* < 0.05). Rapamycin partially restored the expression of the Ref(2)P autophagy protein, but did not alter HO-1 levels in the rotenone/Repo;cnccRNAi + Rapa group compared with the rotenone/Repo;cnccRNAi + DMSO group (* *p* < 0.05).

## Data Availability

The data presented in this study are available on request from the corresponding author.

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
