# Peer review of "Activation of Nrf2 in Astrocytes Suppressed PD-Like Phenotypes via Antioxidant and Autophagy Pathways in Rat and Drosophila Models"

_cells, 2021, doi:10.3390/cells10081850_

Round 1

Reviewer 1 Report

  1. The rationale for only the inclusion of male rats should be included.
  2. What is the number of animals per experiment, inclusion and exclusion criteria of animals for certain tests?
  3. Symbols for fig 1A are unrecognizable and indistinguishable. 
  4. The technical replicates from the WB analysis should be included.
  5. Fig 6B HO-1 blots seem the signal was bleached, technical replicates would be helpful.
  6. Statement on the Ethical committee overseeing animal usage in experiments should be included.
  7. Statistical comparative analysis of rat neuronal Nrf2 and astrocytic nrf2 should be considered to illustrate if Glial nrf2 expression was driving the phenotype Fig. 2.

Author Response

  1. The rationale for only the inclusion of male rats should be included.

Answer: There was no significant difference between male and female rats, and in order to maintain the consistency of research conditions, we used the male adult SD rats for all rat experiments. We pointed out this reason in the manuscript.

  1. What is the number of animals per experiment, inclusion and exclusion criteria of animals for certain tests?

Answer: 48 rats were randomly divided into 4 groups (control group, 6-OHDA group, CDDO group and 6-OHDA+CDDOgroup). The apomorphine induced rotational behavior to the contralateral side and the number of rotational behavior more than7 turns/per min or 210 turns/30 min were considered as successful PD model rats. We added this information in the manuscript.

  1. 3. Symbols for fig 1A are unrecognizable and indistinguishable. 

Answer: We have made the correction.

  1. The technical replicates from the WB analysis should be included.

Answer: The experiments of WB were repeated at least three times, and the statistical analysis included all the analysis results together.

  1. Fig 6B HO-1 blots seem the signal was bleached, technical replicates would be helpful.

Answer: Thank you for your suggestion. We performed western blots of every HO-1 test for at least three times and chosed an image with more significant difference in Fig 6B, but the image is a little blurry. We changed another image for western blots of HO-1 in Fig 6B and re-analyzed results more carefully.

  1. Statement on the Ethical committee overseeing animal usage in experiments should be included.

Answer: The Animal Ethics Committee of Soochow University (Suzhou, China) approved the animal protocols on March 2, 2018. All experiments were designed according to the guidelines of animal experimental center of Soochow University. We added this information in the manuscript.

  1. Statistical comparative analysis of rat neuronal Nrf2 and astrocytic nrf2 should be considered to illustrate if Glial nrf2 expression was driving the phenotype Fig. 2.

Answer: In our experiments we also used NSE antibody for co-staining of Nrf2 with neurons. We also compared nrf2 level in neurons and astrocyts and observed increased Nrf2 signal in glial cells, the Nrf2 signal in neurons was not changed significantly. So we conducted subsequent experiments to confirm the roles of Nrf2 in glial cells.

Reviewer 2 Report

Guo et al. investigates the role of Nrf2, antioxidants and authophagy in Parkinson's disease. For that, they use two different organism models: rats and flies.

The topic manuscript is potentially interesting. However, I have serious concerns about the experimental methodology used and thus the results obtained. The material and method section needs to be improved and more information provided. The bibliography provided is not sufficient and the discussion a repetition of the manuscript conclusions. Additionally, the quality of the figures is very poor: Figures are small, blurry and hence hard to read and evaluate.

Ethical concerns:

  • No approval number for the rat studies is provided.
  • What is the role of Y.L. in the study? According the manuscript, Y.L. is just a corresponding author and did not contribute to the research or wrote the manuscript. Moreover, Y.L. corresponding information is not provided...

Main experimental concerns:

  • Microscope (confocal?) images in figures 1 and 2 have obvious different background signals between treatments. Therefore their quantification and interpretation are compromised. All images should be acquired with the same microscope settings and minimising background signals.
  • Knockdown of Nrf2 (Fig.6) must be verified by qPCR and there is an obvious problem with the signal acquisition of the Western blot of HO-1

Material and methods concerns.

  • The food of the rats and flies must be described.
  • Several antibodies used are not described in the method section.
  • Sex and age of the flies in the drug treatment and climbing test need to be included.
  • 20-25 female flies is a low n number for the standards of the field. Also, a Logrank test or similar should be used to confirm the significance of the survival curves. How authors obtained a half-survival value with error bars? It should be a single point.
  • What was the high of the test line on the climbing test?
  • Which buffer was used to homogenise the brain tissues of the rats? Also, the protocol of homogenisation needs to be described.
  • Authors should describe the reagents/machines better. For instance, what confocal microscope was used? What specific reagent what used to detect the signal of the Western blots and in which machine? About the SDS-pages: What %? Commercial gels or handmade? PDVF membranes from which company? Hoechst from and what concentration (molar or mg/ml)? And many others.

Please define 6-OHDA , CDDO-Me in the main text and in the abstract. Please include always as space between the number and the unit, i.e. 3 mM.

Author Response

  1. The material and method section needs to be improved and more information provided. The bibliography provided is not sufficient and the discussion a repetition of the manuscript conclusions.

Answer: We improved these in the manuscript as suggested.

  1. No approval number for the rat studies is provided.

Answer: 48 rats were randomly divided into 4 groups (control group, 6-OHDA group, CDDO group and 6-OHDA+CDDOgroup) and all experiments were designed according to the guidelines of animal experimental center of Soochow University. We added this information in the manuscript.

  1. What is the role of Y.L. in the study? According the manuscript, Y.L. is just a corresponding author and did not contribute to the research or wrote the manuscript. Moreover, Y.L. corresponding information is not provided the manuscript.

Answer: We have completed all author information in the manuscript.

  1. Microscope (confocal?) images in figures 1 and 2 have obvious different background signals between treatments. Therefore their quantification and interpretation are compromised. All images should be acquired with the same microscope settings and minimising background signals.

Answer: We took pictures by fluorescence microscope using the same setup. Due to the difference of the primary antibody in the staining, some images had a higher background. We adjusted the background of the images and redid the pictures in figures.

5.Knockdown of Nrf2 (Fig.6) must be verified by qPCR and there is an obvious problem with the signal acquisition of the Western blot of HO-1

Answer: For all knockdown experiments by UAS-RNAi, we need to performe qRT-PCR for analysis of knockdown effects for the UAS-RNAi fly lines. In experiments of knockdown of cncc (fly's Nrf2orthologue) in glial cells by Repo-GLA4; UAS-cnccRNAi, the qRT-PCR primers for cncc were 5′-GAGGTGGAAATCGGAGATGA-3′ and 5′-CTGCTTGTAGAGCACCTCAGC-3′. We added this information in the manuscript.

6.The food of the rats and flies must be described.

Answer: The 1 liter standard medium consisting of 90 g Corn powder, 75 g sucrose, 36 g dry yeast, 10 g agar, 4.5ml of propionic acid, and 2 g nipagin. We added this information in the manuscript. The food of the rats was provided by animal experimental center of Soochow University. We added this information in the manuscript.

7.Several antibodies used are not described in the method section.

Answer: Thank you for pointing out and we added information in the manuscript.

  1. Sex and age of the flies in the drug treatment and climbing test need to be included.

Answer: In order to maintain the consistency of research conditions, the female flies were used for all experiments. For drug treatment, the drug were added to the fly food vials start from collection of newly eclosion files. For climbing test, the tests were conducted weekly for whole survival time.

9.20-25 female flies is a low n number for the standards of the field. Also, a Logrank test or similar should be used to confirm the significance of the survival curves. How authors obtained a half-survival value with error bars? It should be a single point.

Answer: 20-25 files were for each food vial and 4 parallel vials (total 80-100files) were performed for each experiments group. The Log-rank test was used to analyzesurvival curves and the half survival time was analyzed for a special single point. We completed this information in the manuscript.

10.What was the high of the test line on the climbing test?

Answer: The test line was 10 cm above the bottom in test tube of 15 cm high.

  1. Which buffer was used to homogenise the brain tissues of the rats? Also, the protocol of homogenisation needs to be described.

Answer: The RIPA Lysis Buffer (Beyotime, 50 mM Tris-HCl, pH7.4, 150 mM NaCl, 1% Triton X-100,1% sodium deoxycholate,0.1% SDS )with protease inhibitor PMSF (Beyotime, RIPA: PMSF=100:1). We added this information in the manuscript.

  1. Authors should describe the reagents/machines better. For instance, what confocal microscope was used? What specific reagent what used to detect the signal of the Western blots and in which machine? About the SDS-pages: What %? Commercial gels or handmade? PDVF membranes from which company? Hoechst from and what concentration (molar or mg/ml)? And many others.

Answer: We have made all corrections as suggested, thank you!

  1. Please define 6-OHDA , CDDO-Me in the main text and in the abstract. Please include always as space between the number and the unit, i.e. 3 mM.

Answer: We have made all corrections as suggested.

Reviewer 3 Report

In the manuscript, Qing Guo and co-workers investigated the oxidative stress and autophagy pathways in models of PD. Although the work is of interest, some issues need to be addressed.

The authors state that “The protein levels of Nrf2 and GCLC increased in both the substantia nigra and striatum in the 6-OHDA group compared with those in the control group.” According to Fig. 2C, the Nrf2 level in the substantia nigra does not differ significantly between the 6-OHDA and the control group.    

The authors investigate the autophagy pathway. Is this only macroautophagy or include chaperon-mediated autophagy as well? What type of inhibitor is 3-MA (early or late stage)? The role of beclin-1 levels in autophagy should be discussed in more details.

Materials and methods, especially the description of Western blotting, is not suitable. Instead of actin, tubulin was used as a loading control, many antibodies are not mentioned in this section.

The quality of figures should be improved, especially that of Fig. 1A, Fig. 3, Fig. 4A and B.

Information is missing on page 13 (indicated by yellow).

Author Response

  1. The authors state that “The protein levels of Nrf2 and GCLC increased in both the substantia nigra and striatum in the 6-OHDA group compared with those in the control group.” According to Fig. 2C, the Nrf2 level in the substantia nigra does not differ significantly between the 6-OHDA and the control group.    

Answer: Yes, the result showed Nrf2 level in the substantia nigra was not significant increased in 6-OHDA group compared with control.  We corrected it in the manuscript.

  1. The authors investigate the autophagy pathway. Is this only macroautophagy or include chaperon-mediated autophagy as well? What type of inhibitor is 3-MA (early or late stage)? The role of beclin-1 levels in autophagy should be discussed in more details.

Answer: 3-MA is a classic autophagy inhibitor that mainly inhibits autophagosome formation and development.Literatures show that 3-methyladenine is a Phosphatidylinositol 3-kinase inhibitors and involvement of beclin-1/Vps34 (class III PI3K) autophagic pathway, therefore we measured the level of beclin-1 in our experiment. We also measured the level of LC3 meanwhile and the results showd changes of both levels are similar, so we present the results of beclin-1 in Fig2. We consider that macroautophagy was mainly affected autophagy pathway and included the stages of autophagosome formation in our experiment. We also think Mitophagy was involved and we need to do more experiments to test it in the future.

  1. Materials and methods, especially the description of Western blotting, is not suitable. Instead of actin, tubulin was used as a loading control, many antibodies are not mentioned in this section.

Answer: We have made all corrections as suggested, thank you!

  1. The quality of figures should be improved, especially that of Fig. 1A, Fig. 3, Fig. 4A and B.

Answer: We reproduce the picture Figure 1A. However, the pictures in Fig.3 A, Fig. 4A and B are made by statistical software, so it is difficult for us to reproduce them more clearly. Thank you for your advice.

5.Information is missing on page 13 (indicated by yellow).

Answer: We have made corrections as suggested, thank you!

Reviewer 4 Report

The paper titled: Activation of Nrf2 in astrocytes suppressed PD-like pheno- 
types via antioxidant and autophagy pathways in rat and Drosophila models by Guo et al. is interesting. Nevertheless same point need to be addressed:

material and methods:

-the animal groups lack of In the saline+CDDO group. Need to be added to the experimental design.

-the experimantal design of Drosophila is poorly described.

-The metods to obtain Knockdow Nrf2 377 in glial cells in Drosophila model is missing.

Results:

TH staining need to be investigate also in striatum not only in nigra.

Figure 2 A and Figure 2B the images are too small. Authors 'd provide an higher magnification of picture to better visualize the NRF2 traslocation.

Why do Authors measured Beclin 1 and not LC3 to investigate autophagy pathway?

Discussion

NRF2 is implicated in mitophagy and mithocondriogenesis Authors 'd discuss the role of NRF2 in their model respect to mitophagy and mithocondriogenesis taking to account the oxidative stress present in theu models.

Author Response

1.the animal groups lack of In the saline+CDDO group. Need to be added to the experimental design.

Answer: We had saline+CDDO group in the experimental design for rat experiments and added the information of this group in the method. There was no significant difference between saline+CDDO and control group, and we mainly focused on the PD model and the changes after administration of CDDO-Me, so we showed the experimental results of control, 6-OHDA and6-OHDA+CDDO groups in the manuscript.

  1. the experimantal design of Drosophila is poorly described.

Answer: We have made corrections as suggested.

3.The methods to obtain Knockdow Nrf2 377 in glial cells in Drosophila model is missing.

Answer: For overexpression and  knockdown of drosophila Nrf2 orthologue "cncc" we used the hybridization of UAS-cncc or UAS-cnccRNAi fly lines with elav-Gal4(expression of target genes in neurons) and Repo-GLA4(expression of target genes in glial cells). We added this information in the manuscript.

  1. TH staining need to be investigate also in striatum not only in nigra.

Answer: We have conducted TH staining in both the substantia nigra and striatum. The cell body of dopamine neurons are mainly located in the substantia nigra, and the fibrous terminal of dopamine neuron are located in the striatum. Therefore we analyzed the number of cell body of survival dopamine neurons in the substantia nigra.

  1. Figure 2 A and Figure 2B the images are too small. Authors 'd provide an higher magnification of picture to better visualize the NRF2 traslocation.

Answer: The pictures were taken using a fluorescence microscope, thus the pictures of Figure 2 A and 2B are the highest magnification in our fluorescence microscope.

6.Why do Authors measured Beclin 1 and not LC3 to investigate autophagy pathway?

Answer: beclin-1/Vps34 (class III PI3K) is involved in PI3K autophagic pathway and autophagosome formation in whole autophagy. We also measured the level of LC3 meanwhile and the results showd changes of both levels are similar, since 3-MA is a PI3kinase inhibitors and involved in PI3K autophagic pathway, so we present the results of beclin-1 in Fig2. For more consideration, we also think LC3 level is a better signal for autophagy here, therefore we present the results of LC3 level in Fig2D now.

  1. NRF2 is implicated in mitophagy and mithocondriogenesis Authors 'd discuss the role of NRF2 in their model respect to mitophagy and mithocondriogenesis taking to account the oxidative stress present in theu models.

Answer: We also consider mitophagy was involved in our models and add more information of mitophagy and mithocondriogenesis in the discussion. We also think we need to do more experiments to test it in the future.

Round 2

Reviewer 2 Report

I appreciate the effort of the authors to improve the manuscript. The methodology section has been certainly improved. However, I have concerns about the manuscript.

  • Images in Figure 2 (have obvious different background signals between treatments. Therefore their quantification and interpretation are compromised. I understand that the antibody can give slightly different background, but this case is beyond that and the quality of the experiment hence low. The experiment need to be repeated and more convincing data presented.
  • How have Beclin blot become now LC3 blot?
  • Figure quality must be improved: low resolution and hard to read.
  • The bibliography provided is not enough and the discussion a repetition of the manuscript results.
  • Why  are the results about knockdown not shown? Please include them in the manuscript. Where the knockdown statistically significant?
  • Why panel C has been removed from Figure 1?

Author Response

Images in Figure 2 (have obvious different background signals between treatments. Therefore their quantification and interpretation are compromised. I understand that the antibody can give slightly different background, but this case is beyond that and the quality of the experiment hence low. The experiment need to be repeated and more convincing data presented.

Answer: The experiments of immunofluorescent staining were repeated and we made new pictures for Figure 2.

How have Beclin blot become now LC3 blot?

Answer: beclin-1/Vps34 (class III PI3K) is involved in PI3K autophagic pathway and autophagosome formation in whole autophagy. In our experiment we also measured the level of LC3 meanwhile and the results showd changes of both levels are similar, since 3-MA is a PI3kinase inhibitors and involved in PI3K autophagic pathway, so we present the results of beclin-1 in Fig2 before. For more consideration, we think LC3 level is a better signal for autophagy here, therefore we present the results of LC3 level in Fig2D now.

Figure quality must be improved: low resolution and hard to read.

Answer: We made new figures in GFAP and Nrf2 immunofluorescent staining.

The bibliography provided is not enough and the discussion a repetition of the manuscript results.

Answer: We added more information in the discussion of manuscript.

Why are the results about knockdown not shown? Please include them in the manuscript. Where the knockdown statistically significant?

Answer: We added this information in the manuscript.

Why panel C has been removed from Figure 1?

Answer: We took new images in panel C of Figure 1.

Reviewer 3 Report

The authors have answered my questions.

Author Response

Thank you very much!

Reviewer 4 Report

The Authors have provided to answer to the questions.

The manuscript can be accepted in the present form

Author Response

Thank you very much!

Round 3

Reviewer 2 Report

n/a